# Diversity and plasticity in Rab GTPase nucleotide release mechanism has consequences for Rab activation and inactivation

Lars Langemeyer[1], Ricardo Nunes Bastos[1], Yiying Cai[2], Aymelt Itzen[3], Karin M Reinisch[2], Francis A Barr[1]*

[1]Department of Biochemistry, University of Oxford, Oxford, United Kingdom; [2]Department of Cell Biology, Yale University School of Medicine, New Haven, United States; [3]Center for Integrated Protein Science Munich, Technische Universität München, Munich, Germany

**Abstract** Ras superfamily GTPase activation and inactivation occur by canonical nucleotide exchange and GTP hydrolysis mechanisms. Despite conservation of active-site residues, the Ras-related Rab GTPase activation pathway differs from Ras and between different Rabs. Analysis of DENND1-Rab35, Rabex-Rab5, TRAPP-Rab1 and DrrA-Rab1 suggests Rabs have the potential for activation by distinct GDP-release pathways. Conserved active-site residues in the Rab switch II region stabilising the nucleotide-free form differentiate these pathways. For DENND1-Rab35 and DrrA-Rab1 the Rab active-site glutamine, often mutated to create constitutively active forms, is involved in GEF mediated GDP-release. By contrast, in Rab5 the switch II aspartate is required for Rabex mediated GDP-release. Furthermore, Rab1 switch II glutamine mutants refractory to activation by DrrA can be activated by TRAPP, showing that a single Rab can be activated by more than one mechanistically distinct GDP-release pathway. These findings highlight plasticity in the activation mechanisms of closely related Rab GTPases.

*For correspondence: francis. barr@bioch.ox.ac.uk

**Competing interests:** The authors declare that no competing interests exist.

**Reviewing editor**: Suzanne R Pfeffer, Stanford University, United States

## Introduction

Rabs form an important and highly conserved subfamily of Ras-related GTPases that play essential roles in controlling membrane trafficking between the organelles of eukaryotic cells (*Zerial and McBride, 2001*; *Pfeffer and Aivazian, 2004*). Specific regulators controlling nucleotide exchange and hydrolysis promote kinetic proofreading of vesicle and target organelle membrane surfaces by Rab GTPases, and therefore permit accumulation of active Rabs only at the required sites (*Barr and Lambright, 2010*; *Barr, 2013*). The mechanistic details of how such regulators control Rab activation is therefore important for understanding the regulation of membrane identity and vesicle transport. Activation of Ras superfamily GTPases is thought to proceed by a general nucleotide exchange mechanism (*Bos et al., 2007*). In the Ras-SOS GEF (guanine nucleotide exchange factor) complex the Ras P-loop lysine interacts with a conserved glutamate intrinsic to the Ras active site switch II region, thereby stabilising the GEF bound nucleotide-free form of the GTPase (*Boriack-Sjodin et al., 1998*). Mutation of this glutamate therefore reduces GEF-stimulated GDP-release, and compromises Ras activation (*Gasper et al., 2008*). Because of the high degree of sequence conservation in the Ras superfamily and Rab subfamily (*Klopper et al., 2012*; *Rojas et al., 2012*), this mechanism might be expected to be the same in the Rab subfamily of GTPases. However, at odds with this simple idea mutation of the conserved Rab switch II glutamate residue to alanine has little effect on the rate of GEF-mediated nucleotide exchange (*Gasper et al., 2008*).

**eLife digest** The 70 or so members of the Rab subfamily of proteins perform a wide range of important tasks inside cells. A Rab protein is always bound to another molecule, which determines whether it is inactive or active. Binding to a molecule called GDP makes the Rab protein inactive, while binding to GTP makes it active. Proteins called guanine nucleotide exchange factors, or GEFs for short, activate the Rab protein by promoting the release of GDP and the binding of GTP. Other proteins—known as GAPs—lead to the inactivation of the Rab protein. Together these proteins form a molecular switch that can be turned on and off.

The Rab subfamily of proteins is part of the large Ras superfamily, and all members of this superfamily are activated and inactivated in a similar way, with the binding and unbinding of GDP and GTP taking place at a structure called the G-domain. The fact that the detailed structure of this domain (at the level of individual amino acids) has been conserved over evolution is often taken as an indication that its mechanism has also been conserved. Langemeyer et al. have now tested this assumption with four different types of GEFs—three from humans and one from the bacteria that cause Listeria—and found that the story is more complicated than expected.

The experiments showed that different amino acids in the active site of the Rab protein are involved when the GEFs mediate the release of the GDP during the activation process. For example, the amino acid glutamine is involved when the Listeria GEF and one of the human GEFs activate the protein, whereas a different amino acid—aspartate—is involved when one of the other human GEFs is responsible for the activation. Using this information, Langemeyer et al. create a human Rab protein that cannot be activated by the GEF from the bacteria that cause Listeria, but can still be activated by its normal human GEF.

By showing that different Rab proteins are activated by different mechanisms, and that a single Rab protein can be activated by more than one mechanism, the work of Langemeyer et al. clearly illustrates the on-going ability of evolution to surprise researchers.

In addition to these potential differences between Ras and Rab activation, the mechanism of Rab inactivation by GTPase activating proteins (GAPs) diverges in key details from Ras. In Ras the conserved switch II glutamine 61 and an arginine residue contributed into the Ras active site by the GAP act together to promote GTP hydrolysis (*Ahmadian et al., 1997*; *Scheffzek et al., 1997*). Mutation of either residue therefore prevents GTP hydrolysis. This has great biological relevance since the Ras switch II glutamine is frequently mutated in cancers creating a constitutively active oncogenic form of the protein. In Rabs, although the switch II glutamine is conserved, crystal structures of Rab33 and Rab1 with TBC domain Rab GAPs, Gyp1p and TBC1D20, respectively, reveal that it does not play a direct role in GTP hydrolysis (*Pan et al., 2006*; *Gavriljuk et al., 2012*). Instead, the GAP contributes both arginine and glutamine residues important for catalysis to the Rab activate site (*Pan et al., 2006*; *Gavriljuk et al., 2012*).

Therefore, despite the high level of sequence conservation in the key switch regions of Ras and Rab family members, the mechanisms of activation and inactivation may differ between Ras and Rabs. In other words shared sequence cannot be assumed to imply shared mechanism. We therefore investigated the role of conserved Rab switch II active site residues in GEF-mediated activation to obtain insight into the function and reasons for their conservation. This analysis revealed that Rab activation diverges from the canonical Ras pathway in key details. Furthermore, we find that even within the Rab family different activation mechanisms are used by different Rabs.

## Results

### Differences in switch II interactions in Rab-GEF complexes

Inspection of the Ras-SOS GEF complex shows that the Ras P-loop lysine interacts with a conserved glutamate intrinsic to the active site switch II region (*Figure 1A*). Scrutiny of DENND1-Rab35 (*Wu et al., 2011*) and Rab1-DrrA (*Schoebel et al., 2009*; *Suh et al., 2010*) Rab-GEF complex crystal structures suggests divergence from the Ras activation pathway. In both DENND1 and DrrA complexes, the Rab P-loop lysine 21 interacts with the switch II glutamine 67 and aspartate 63 of the target GTPase

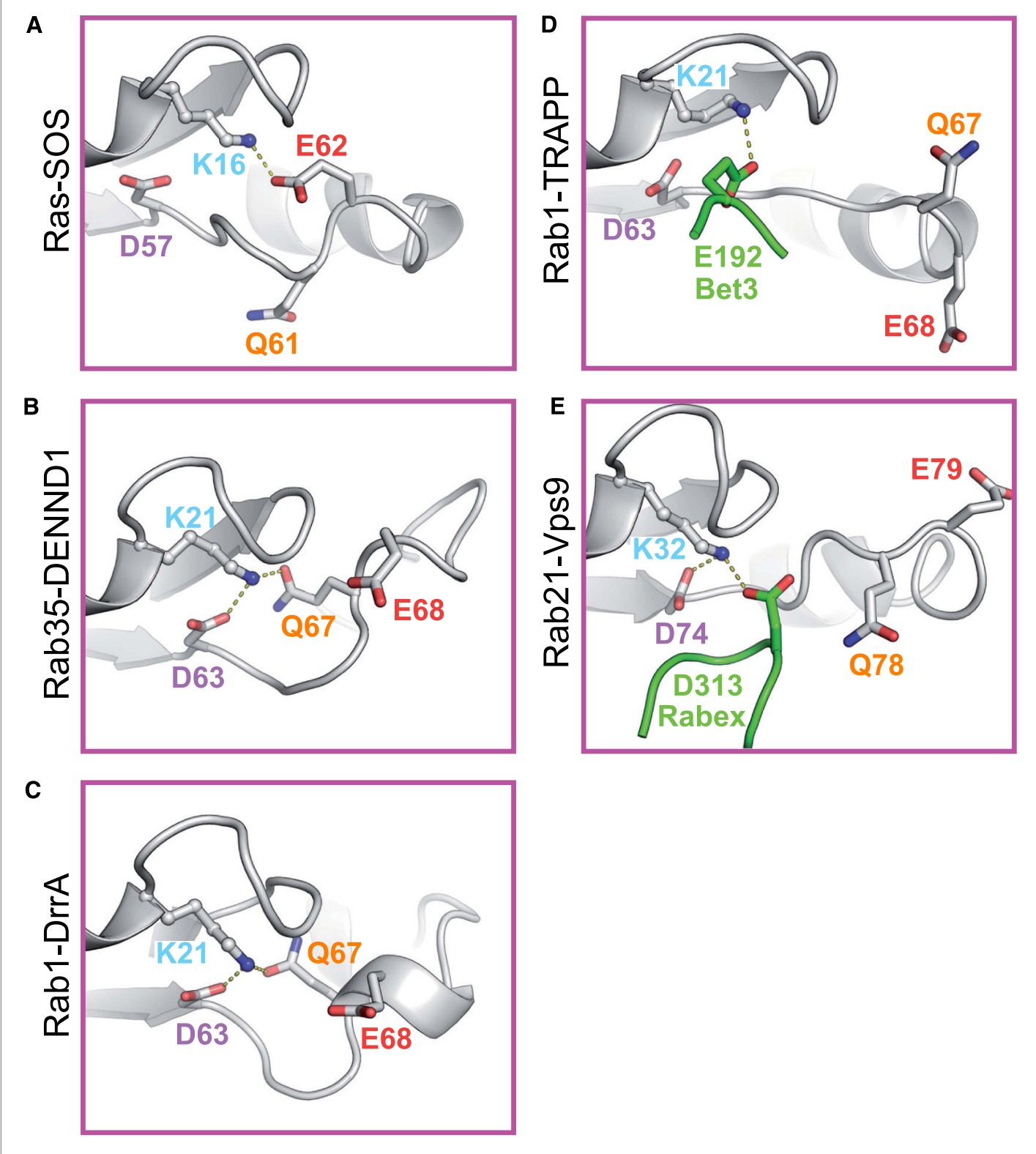

**Figure 1**. Role of switch II residues in Rab GEF complexes. (**A**) The crystal structure of Ras with its exchange factor SOS highlighting the interaction of the Ras P-loop lysine 16 with the Ras switch II glutamate 62. Dotted yellow lines indicate potential Ras P-loop lysine interactions. (**B**) Portions of Rab structures from Rab–RabGEF complex crystal structures are shown for Rab35-DENND1 and (**C**) Rab1-DrrA. (**D**) Ypt1 (budding yeast Rab1) TRAPP and (**E**) Rab21-Vps9/Rabex complexes are shown. The Rab is indicated in grey while the GEF is depicted in green. Switch II residues are coloured according to their position for ease of reference.

(*Figure 1B,C*). Examination of other Rab GEF complexes provides further support for the view that Rab activation differs from Ras. In the case of Rab1 bound to TRAPP there is no interaction of the Rab P-loop lysine with switch II residues (*Figure 1D*). Instead, a glutamate at position 192 is contributed to the Rab active site by the C-terminal extension of the TRAPP Bet3 subunit (*Cai et al., 2008*; *Chin et al., 2009*; *Figure 1D*). This could be considered mimicry of the Rab switch II region glutamate by the GEF. For the Rab5 family GTPase Rab21 in complex with the Vps9 domain of the Rabex exchange factor, a conserved aspartate 74 in the switch II region of the Rab interacts with the P-loop lysine 32 (*Delprato et al., 2004*; *Delprato and Lambright, 2007*; *Figure 1E*). Analogous to TRAPP an acidic residue is also contributed by Rabex, in this case an aspartate at position 313 (*Figure 1E*). These different modes of Rab P-loop interaction with the switch II region or residues from the GEF suggest distinct GDP-release pathways exist for Rab GTPases.

## Distinct roles for switch II residues in Rab activation

To test this idea a series of GEFs acting on Rab GTPases were analysed. First, the Rab35 GEF DENND1 was compared to the Rab1 GEF DrrA. Rab35 switch II glutamine 67 mutation to alanine (Q67A) greatly reduced DENND1-stimulated nucleotide exchange towards Rab35 (*Figure 2A*). Catalytic efficiency ($k_{cat}/K_m$) was reduced from $2.3 \times 10^4$ $M^{-1}s^{-1}$ similar to previous measurements using wild type Rab35 (*Wu et al., 2011*) to $\sim 7.5 \times 10^2$ $M^{-1}s^{-1}$ for the Q67A mutant. Importantly, the glutamine to alanine mutation had little effect on basal GEF-independent nucleotide exchange. In the case of Rab35 there is a specific requirement for glutamine at this position since alteration of the switch II

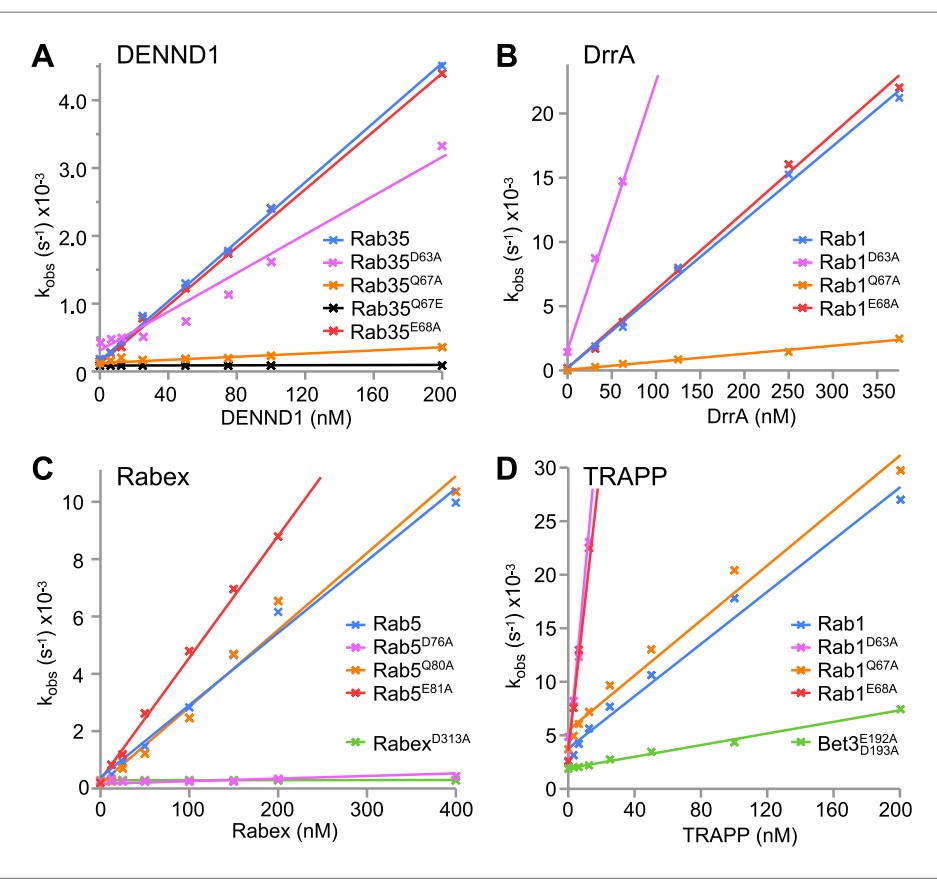

**Figure 2**. Distinct roles for switch II residues in GEF-mediated nucleotide exchange independent of the basal nucleotide release pathway. (**A**) Initial rates of nucleotide exchange as a function of GEF concentration are plotted for Rab35-DENND1, (**B**) Rab1-DrrA, (**C**) Rab5-Rabex and (**D**) Rab1-TRAPP. Wild type and mutant Rabs were used as indicated; curves are colour coded as in *Figure 1* according to the position in the switch II region predicted to be important for GEF-mediated nucleotide release. Wild type full-length GEFs were used for DENND1, DrrA, Rabex and TRAPP, as well as the Rabex D313A mutant, and the TRAPP Bet3 E192A/D193A mutant.

glutamine to glutamate resulted in a form of Rab35 Q67E that was not activated by DENND1 (*Figure 2A*). In addition, mutation of the switch II glutamate 68 (E68A) required for nucleotide exchange in the canonical Ras activation pathway had little effect on DENND1-stimulated nucleotide exchange by Rab35 (*Figure 2A*). Similarly, the Q67A switch II mutation greatly reduced DrrA-mediated nucleotide exchange towards Rab1 from a $k_{cat}/K_m$ of $6.0 \times 10^4$ to $5.3 \times 10^3$ $M^{-1}s^{-1}$ (*Figure 2B*). Mutation of the switch II glutamate (E68A) in Rab1 had little effect on DrrA-mediated nucleotide exchange, $k_{cat}/K_m$ $6.0 \times 10^4$ $M^{-1}s^{-1}$ (*Figure 2B*). Removal of the switch II aspartate 63 contacting the P-loop lysine residue resulted in increased basal release for both Rab35 and Rab1 (*Figure 2A,B*), consistent with its role in nucleotide binding. For DENND1 this substitution resulted in a small reduction in $k_{cat}/K_m$ to $1.7 \times 10^4$ $M^{-1}s^{-1}$ (*Figure 2A*), while for DrrA $k_{cat}/K_m$ increased over threefold to $2 \times 10^5$ $M^{-1}s^{-1}$ (*Figure 2B*). These results suggest that the switch II glutamine in these Rabs plays a crucial and previously unsuspected role in GDP-release during Rab activation. By contrast, the switch II aspartate contributes to nucleotide binding and therefore limits GDP-release (*Pai et al., 1989*; *John et al., 1993*).

The structure of the Rab5 family member Rab21 with its Vps9 domain GEF Rabex suggested that in this case the conserved glutamine is not in a position to promote nucleotide exchange (*Figure 1E*). Instead, the structure suggests an alternative pathway where the Rab P-loop lysine interacts with the Rab switch II aspartate and an aspartate finger residue D313 in Rabex. As reported previously, mutation of the Rabex aspartate to alanine D313A abolished the activity of the GEF (*Figure 2C*; *Delprato et al., 2004*). In support of the role of the Rab switch II aspartate in this alternative exchange pathway, mutation of the Rab5 conserved aspartate D76A greatly reduced Rabex-mediated nucleotide exchange (*Figure 2C*). Because the numbering of Rab5c is increased by two amino acids with respect to the Rab21, the same residue is D74 in the Rab21 structure (*Figure 1E*). Catalytic efficiency was reduced from $2.5 \times 10^4$ $M^{-1}s^{-1}$ in agreement with previous measurements using wild type Rab5 (*Delprato et al., 2004*; *Delprato and Lambright, 2007*) to ~$2.5 \times 10^2$ $M^{-1}s^{-1}$ for the D76A mutant. As expected from the crystal structure of the Rab21-Rabex complex (*Figure 1E*), the Q80A mutation (Q78 in *Figure 1E*) had little effect on the activity of Rabex towards Rab5 (*Figure 2C*). Interestingly, mutation of Rab5 switch II glutamate 81 (Rab21 E79 in *Figure 1E*) resulted in an increase in the rate of Rabex-stimulated GDP-release from a $k_{cat}/K_m$ of $2.5 \times 10^4$ to $4.6 \times 10^4$ $M^{-1}s^{-1}$ (*Figure 2C*). This increase is possibly due to removal of the negative charge on the switch II favouring entry of Rabex aspartate 313 into the vicinity of the P-loop lysine. Rab5 activation by Rabex therefore proceeds via an alternative pathway in which the conserved switch II aspartate interacts with the P-loop lysine to promote GDP-release. This is different to Rab35-DENND1 or Ras-SOS where the switch II glutamine or glutamate fulfil an equivalent role.

## Analysis of TRAPP and DrrA reveals plasticity in Rab1 activation

Both TRAPP and DrrA can activate Rab1 family GTPases, however the crystal structures of these Rab-GEF complexes reveal differences in the interaction network around the P-loop lysine. For Rab1-DrrA complexes, the Rab1 P-loop lysine contacts the switch II aspartate 63 and glutamine 67 residues (*Figure 1C*). This is different to the structure of Rab1 with its longin domain GEF TRAPP where there are no obvious contacts between switch II and the P-loop lysine 21 (*Figure 1D*). Instead, the structure suggests an alternative pathway where the Rab1 P-loop lysine interacts with an acidic glutamate finger residue E192 provided by the Bet3 subunit of the TRAPP GEF complex. The Rab1 switch II glutamine mutation is therefore predicted to have no effect on TRAPP mediated nucleotide exchange. In agreement with this idea, the switch II glutamine mutation Q67A that greatly reduced DrrA-mediated nucleotide exchange towards Rab1 (*Figure 2B*) had little effect on the activity of budding yeast TRAPP towards Rab1/Ypt1 (*Figure 2D*). As for DrrA, removal of the Rab1 switch II aspartate 63 contacting the P-loop lysine residue resulted in increased basal and TRAPP-stimulated GDP-release (*Figure 2D*). Interestingly, mutation of switch II glutamate 68 increased the rate of TRAPP-stimulated GDP-release over 10-fold from a $k_{cat}/K_m$ of $1.3 \times 10^5$ $M^{-1}s^{-1}$. This was possibly due to removal of the negative charge on the switch II favouring entry of Bet3 glutamate 192 into the vicinity of the P-loop lysine. To confirm the requirement for the Bet3 acidic C-terminal region, TRAPP complexes containing a Bet3 E192A/D193A mutant were produced. These showed a fourfold reduction in GEF activity towards Rab1 (*Figure 2C*). This is similar to the result obtained with Rabex, which also inserts an acidic residue to coordinate the P-loop lysine during GDP-release. Rab1 activation by TRAPP therefore proceeds via a pathway in which the P-loop lysine

does not interact with switch II, but is stabilised by an acidic residue from the GEF thereby promoting GDP-release.

Together, these results define previously unrecognised pathways for activation of a Ras superfamily GTPase that are discrete from the switch II glutamate dependent pathway followed by Ras. Notably, the results obtained with Rab1 also reveal that a single Ras superfamily GTPase has the potential to follow more than one activation pathway depending on the GEF it is coupling with.

## Differential requirements for the switch II glutamine in GTP hydrolysis

Since the switch II glutamine appears to be an important element in the activation pathway of Rab proteins by GEFs, we also wanted to analyse the significance of its involvement in GAP-mediated Rab-inactivation. Therefore, the role of the switch II glutamine in the GTP hydrolysis reaction leading to Rab inactivation was examined. Crystal structures of Rab33 and Rab1 with TBC domain Rab GAPs, Gyp1p and TBC1D20, respectively, show that the GAP rather than the cognate GTPase contributes the glutamine residue important for nucleotide hydrolysis (*Pan et al., 2006*; *Gavriljuk et al., 2012*). In these cases the only contribution of the switch II glutamine appears to be in contacting the peptide backbone of the GAP. Therefore it may stabilise the Rab-GAP complex rather than playing a direct role in catalysis. The TBC domain Rab GAPs acting on Rab35, Rab5 and Rab1 were then analysed (*Haas et al., 2005*, *2007*; *Fuchs et al., 2007*). Switch II glutamine mutation of Rab35 and Rab1 greatly reduced GTP hydrolysis stimulated by TBC1D10A and TBC1D20, respectively, but had less than ~1.5-fold effect on basal GTP hydrolysis (*Figure 3A,B*). By contrast, the switch II glutamine mutation had no effect on the activity of RUTBC3 towards Rab5, but caused a greater than fivefold reduction in basal GTP hydrolysis (*Figure 3C*). Consequently, depending on the Rab, the switch II glutamine is important for Rab inactivation by GAP-stimulated or basal hydrolysis reactions.

## A Rab35 switch II glutamine mutant is inactive not dominant active

Rab switch II glutamine mutants have been widely used as dominant active forms. This is best understood for Rab5 where such mutants promote endosome fusion (*Stenmark et al., 1994*; *Rybin et al., 1996*). The results presented here indicate that this may not apply to other Rabs, since the same mutation can interfere with GEF stimulated GDP-release and hence Rab activation. This suggests that in some Rabs switch II glutamine mutations will be uncoupled from their GEF and hence inactive rather than dominant active when expressed in cells. This idea was tested using Rab35 and the plasma membrane to Golgi trafficking pathway with Shiga toxin B as cargo protein. These experiments confirmed that wild type Rab35 supported delivery of cargo to the Golgi (*Figure 4A*, 90 min timepoint), as previously reported (*Fuchs et al., 2007*; *Yoshimura et al., 2010*). Under the same conditions the switch II mutant Rab35[Q67A] did not actively interfere with Shiga toxin transport (*Figure 4A*, 90 min timepoint). This indicates it does not act as a dominant negative inhibitor of transport. However, in the absence of the endogenous protein, Rab35[Q67A] failed to support efficient Shiga toxin transport to the Golgi (*Figure 4B*). Instead, Shiga toxin accumulated in small punctate structures and was not efficiently delivered to the Golgi even after 90 min (*Figure 4B*). This reduced accumulation in the Golgi was not due to altered binding of Shiga toxin to the cell surface, since this was equivalent in all the conditions used (*Figure 4A,B*, 0 min timepoint). Quantitation showed transport efficiency was over 80% for wild type Rab35, whereas Rab35[Q67A] did not rescue transport above the level seen in the absence of Rab35 (*Figure 4C*). Therefore Rab35 switch II glutamine mutants are inactive rather than dominant active.

To test if this loss of function is associated with reduced membrane recruitment, the rate of wild type Rab35 and Q67 switch II mutant recruitment to organelle membranes was investigated using fluorescence recovery after photobleaching (FRAP). In agreement with the GEF activity data, Rab35 was rapidly recruited to membranes while the Rab35 Q67A or Q67L mutants showed only slow recovery (*Figure 4D*). For Rab5 the Q80A/L switch II mutation (Q78 in Rab21, *Figure 1E*) did not alter activation by the GEF Rabex (*Figure 2C*), and accordingly had no effect on the rate of recovery (*Figure 4E*), suggesting it is rapidly recruited to membranes like wild type Rab5. Western blot analysis confirmed that at steady state the membrane bound fractions of wild type and switch II glutamine mutant Rab35 and Rab5 were unaltered (*Figure 4D,E*, inset blots). This suggests that membrane anchoring of the Rabs was not affected by the mutations. Together, these results show that switch II glutamine mutant Rab35[Q67A] is defective for rapid membrane recruitment, and fails to support Shiga toxin trafficking to the Golgi.

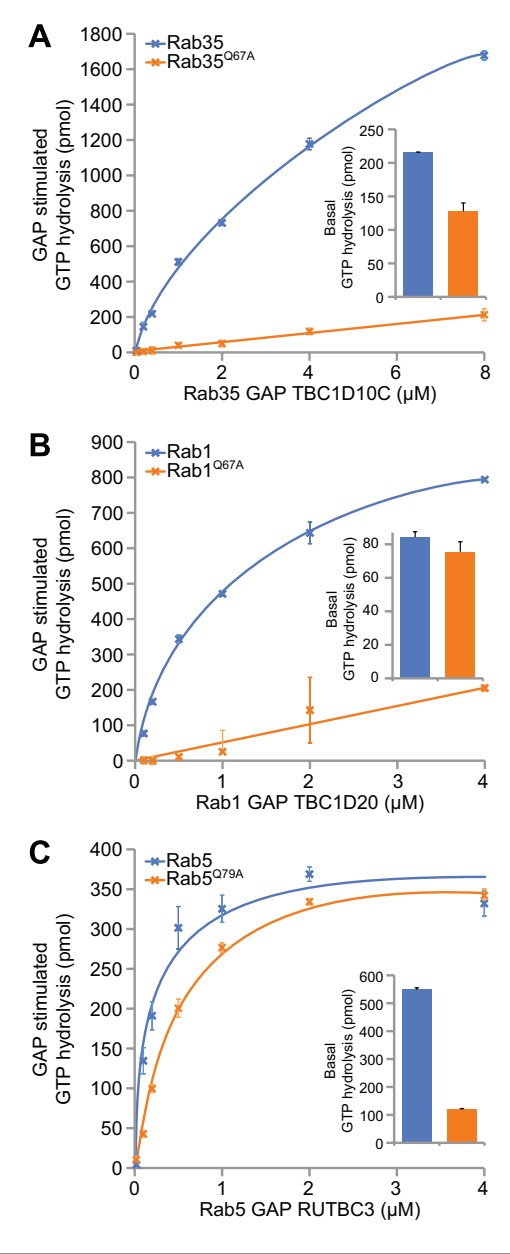

**Figure 3**. Differential requirements for the Rab switch II glutamine in GAP-stimulated GTP hydrolysis. (**A**) Rab GTP hydrolysis as a function of GAP concentration are plotted for Rab35-TBC1D10, (**B**), and Rab1-TBC1D20 (**C**) Rab5-RUTBC3. Both wild type and switch II glutamine mutant Rabs were used. Basal GTP hydrolysis of purified Rabs in the absence of the cognate GAP is shown in the bar graph insets (mean +/− the deviation from the mean, n = 2).

## Discussion

### Divergence of the Ras and Rab activation cycles

The structural and biochemical analysis presented in this study identifies distinct pathways for Rab activation used by different RabGEF families. These can be divided into two main groups depending on the role played by the switch II region. For the first class of Rab GEFs, exemplified by DrrA and the DENN family member DENND1, GDP-release is promoted by interaction of the P-loop lysine with the conserved glutamine residue intrinsic to switch II (*Figure 5A*). Some details of this mechanism remain unresolved, particularly the question of how charge neutralization of the Rab P-loop lysine occurs remains unclear. The Rab–RabGEF complex structures suggest this is possibly due to the negatively charged switch II aspartate, however mutation of this residue does not reduce GEF mediated GDP-release. Future work, analysing structures of such mutants will therefore be necessary. For the second class of GEFs represented by the Vps9 family member Rabex and TRAPP, the Rab P-loop lysine interacts with a negatively charged residue, either aspartate or glutamate, provided by the GEF and extrinsic to the Rab (*Figure 5A*). This latter group can be further subdivided, since stabilisation of the P-loop lysine in Rab1 bound to TRAPP appears not to involve direct interaction with any residues in switch II. All these cases are different to the general nucleotide exchange mechanism proposed for the Ras family, in which formation of the nucleotide-free form of the GTPase is promoted by interaction of a glutamate intrinsic to the switch II region with the P-loop lysine (*Boriack-Sjodin et al., 1998*; *Bos et al., 2007*; *Gasper et al., 2008*). For Ras, this results in discrete requirements for conserved glutamate and glutamine residues in switch II for GEF-mediated activation and GAP-stimulated inactivation, respectively. However, for DENND1 and DrrA, the switch II glutamine of the target Rabs interacts with the P-loop lysine to promote conversion of the Rab GDP form to a nucleotide-free intermediate that then binds GTP (*Figure 5B*). Mutation of this glutamine therefore reduces the rate of GDP release by the Rab. In addition, the same glutamine interacts with the GAP and aides the GTP hydrolysis reaction converting the GTP to a GDP bound Rab (*Figure 5B*), thereby inactivating the Rab. This results in activation and inactivation processes with a shared requirement for the switch II glutamine residue. The glutamine side chain swings from an 'in' conformation where it contacts the P-loop lysine during nucleotide exchange (activation), to an 'out' conformation where it contacts the GAP during GTP hydrolysis (*Figure 5C*). DENND1, the Rab35 GEF studied here, is a member of the DENN and DENN-related proteins that form the largest family of Rab nucleotide exchange factors (*Yoshimura et al., 2010*; *Barr, 2013*). Therefore, these

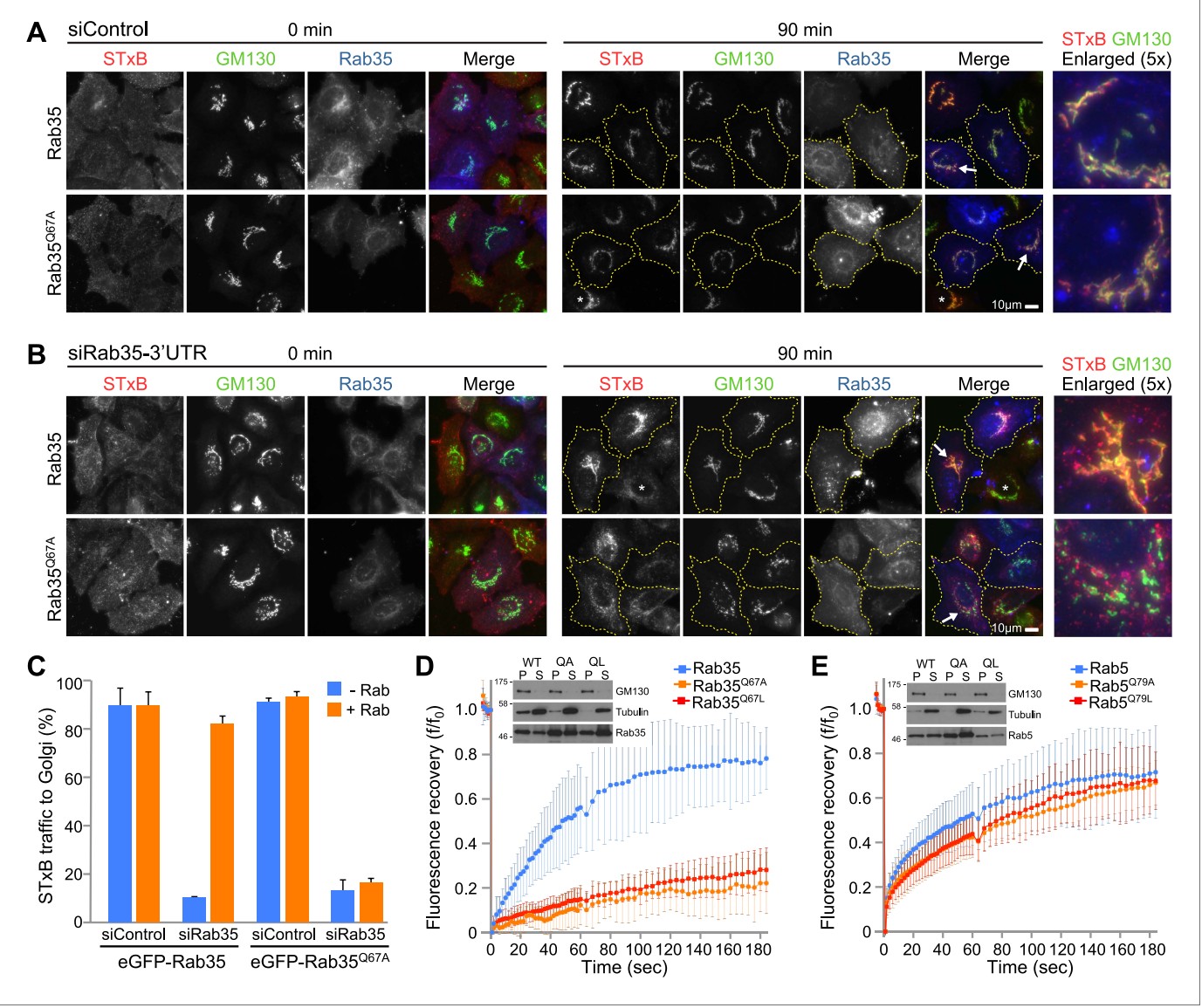

**Figure 4**. The glutamine switch II mutant Rab35 fails to support Shiga toxin transport from the cell surface to the Golgi. (**A**) Shiga toxin B (STxB) uptake assays were performed for 0 and 90 min in HeLa cells expressing either Rab35, or (**B**) the Rab35$^{Q67A}$ mutant. Endogenous Rab35 was depleted using siRNA directed to the 3′-UTR or a mock depletion performed using a non-targeting control duplex. Cells were stained with a GM130 antibody to mark the Golgi. Scale bar is 10 μm. Cells outlined in yellow dotted lines in the 90 min timepoint express GFP-Rab35 or Rab35$^{Q67A}$, and asterisks mark non-transfected cells. Arrowheads mark those cells shown in the enlarged panels to the right. (**C**) Delivery of Shiga toxin into the Golgi was scored and is plotted in the bar graph (mean +/− deviation from the mean, n = 2). (**D**) FRAP experiments were performed on cells expressing wild type and Switch II glutamine mutant Rab35 or (**E**) Rab5 (mean +/− standard deviation from the mean, n = 12). Western blots show the distribution of GFP-Rab35 or GFP-Rab5 in the membrane and cytosol fractions marked by the Golgi membrane protein GM130 and tubulin, respectively.

findings are likely to be broadly applicable to many other Rab GTPases. As a consequence they have wide significance for future studies of membrane trafficking and GTPase regulation in other systems.

## Switch II function in GTP hydrolysis

The findings presented here also indicate that switch II glutamine mutant Rabs will behave differently depending on the nature of their GEF activator and GAP inactivator. For DENN GEF targets such as Rab35, because both GEF-stimulated activation and GAP-mediated inactivation are compromised (*Figures 2A and 3A*), the Rab does not fulfil its cellular purpose (*Figure 4A–C*). For Rab1/Ypt1 activated by TRAPP, switch II mutants slow but do not prevent the GAP-stimulated GTPase reaction

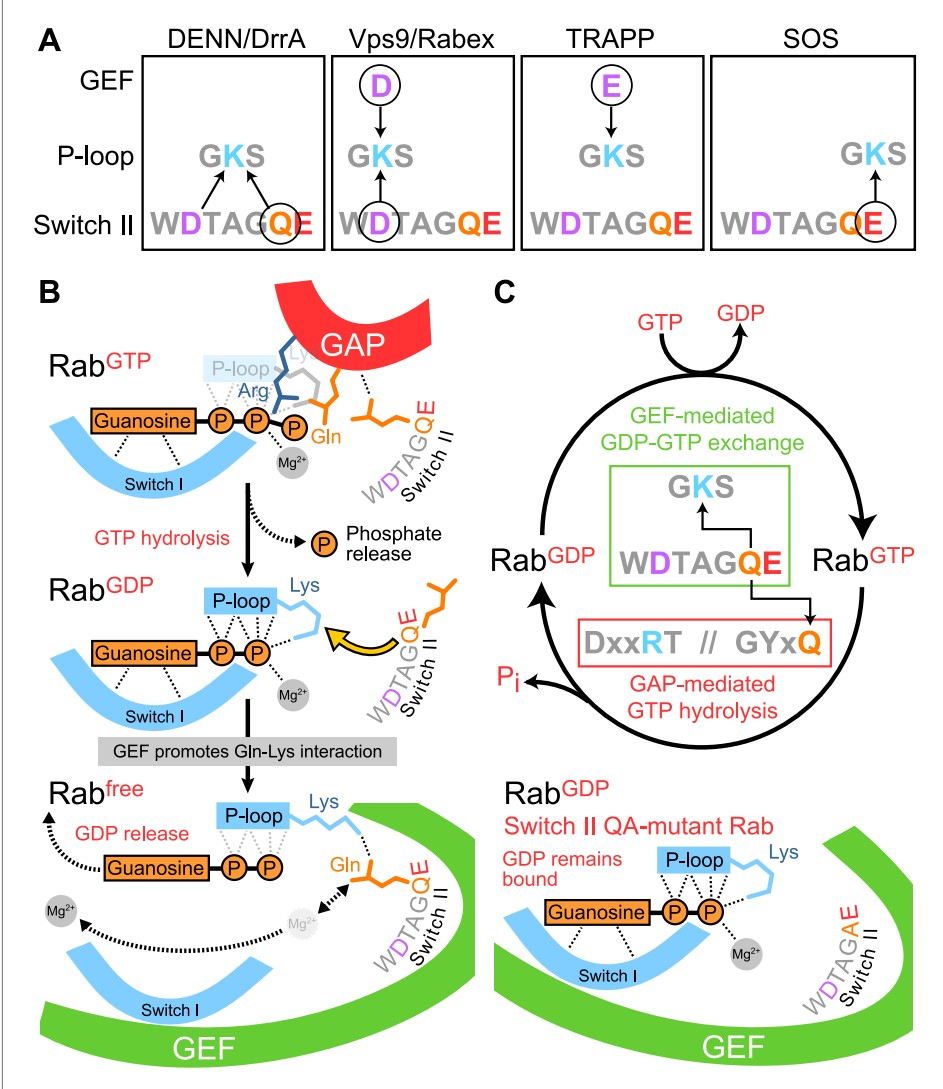

**Figure 5**. Diversity and plasticity in Rab GTPase nucleotide release mechanism. (**A**) A schematic depicting the three different P-loop lysine interactions with the Rab switch II region and GEF, or Ras and the GEF SOS. Circled residues are required for GEF-mediated GDP release. (**B**) The switch II glutamine is required for both Rab GAP stimulated GTP hydrolysis, and DrrA or DENN GEF mediated nucleotide exchange reactions. GEF interaction with the GTPase results in distortion of switch I and II regions, and reduced affinity for both the guanosine and terminal phosphate of bound GDP. The switch II glutamine interacts with the P-loop lysine to displace the β-phosphate. This does not occur for switch II Q-mutant Rabs and GDP-release therefore fails. Adapted from Figure 5 of Thomas and Wittinghofer 2007 (***Thomas et al., 2007***). (**C**) A revised Rab GTPase cycle in which GEF-stimulated activation and GAP-mediated inactivation share a common determinant with respect to switch II.

(***Figure 3B***), possibly explaining the subcritical reduction in transport efficiency reported previously (***De Antoni et al., 2002***). In the case of Rab5, switch II glutamine Q79A mutation prolongs the lifetime of the active state by reducing the intrinsic or spontaneous rate of GTP hydrolysis (***Figure 3C***), and as a result promotes endosome fusion (***Stenmark et al., 1994***; ***Rybin et al., 1996***).

Rab5 has served a paradigm for the function of Rab GTPases in membrane traffic (***Zerial and McBride, 2001***), yet as shown here switch II glutamine mutations in other Rabs may not behave in the same fashion as they do in Rab5. For Rab5 the switch II glutamine mutation results in a form that displays near wild type properties in terms of GAP-stimulated GTP hydrolysis, yet strongly compromised basal GTP hydrolysis. Results obtained previously for Rab33B and its GAP RUTBC1 (***Nottingham et al., 2011***) show this is true for at least one other Rab. However, for Rab1 and Rab35 tested here the

same mutation greatly reduces GAP-stimulated GTP hydrolysis. These results have implications for the interpretation of published work on Rab5 regulation. While removal of the Rab5 GAP RUTBC3 results in moderately enlarged endosomes and a delay in endocytic trafficking, expression of Rab5 glutamine mutants causes formation of grossly enlarged vacuolar early endosomes and blocks the pathway at this stage (*Stenmark et al., 1994*; *Haas et al., 2005*). Together these findings indicate that in cells basal hydrolysis is an important determinant of Rab5 lifetime and turnover. This agrees with the idea that the major regulatory determinants of Rab5 levels at endosomal membranes are inputs driving GEF activity such as ubiquitylated cargo and PI-lipids rather than GAP activity (*Del Conte-Zerial et al., 2008*; *Barr, 2013*). By contrast, GAP-stimulated Rab5 turnover may occur in response to specific signals during cell migration and cell adhesion (*Palamidessi et al., 2013*).

### Switch II glutamine mutants: a trap for the unwary

Switch II glutamine mutants have typically been used to trap Rab GTPases in the active state. As shown here this may generate a GTPase refractory to GEF-mediated activation and GAP-stimulated inactivation. The results reported here therefore have implications for the interpretation of many prior studies of Rab GTPase function that have relied on the use of switch II point mutants for individual Rabs or as part of libraries or Rab toolkits (*Ullrich et al., 1996*; *Richardson et al., 1998*; *Clark et al., 2011*; *Dambournet et al., 2011*; *Gallegos et al., 2012*; *Ishida et al., 2012*; *Xiong et al., 2012*). Depending on the nature of the GEF for the GTPase in question, then, the mutation may have additional consequences for GEF stimulated nucleotide exchange. Where this is not known the results of such experiments must be treated with great caution.

### Multiple switch II configurations reveal plasticity in Rab activation

In analysing the role of switch II residues, this work has uncovered plasticity in terms of the different pathways leading to Rab activation. Although all Rabs switch between common GDP and GTP bound states, the intermediates of the nucleotide exchange reaction differ in crucial features. In some cases a single Rab, shown here for Rab1, can follow a different activation pathway depending on the nature of its cognate GEF. Importantly, switch II mutations that compromise one pathway do not necessarily affect other activation routes. Thus, Rab1 switch II glutamine mutants can be activated by TRAPP, yet are refractory to activation by DrrA. These findings raise the possibility that small molecule inhibitors could be developed to target these mechanistically discrete pathways and associated switch II conformations.

## Materials and methods

### Reagents and antibodies

General laboratory chemicals were obtained from Sigma–Aldrich, UK and Fisher Scientific, UK. Commercially available antibodies were used to GM130 (mouse clone 35; BD Biosciences, UK). Secondary antibodies raised in donkey to mouse, rabbit, sheep/goat, and human conjugated to HRP, Alexa-488, Alexa-555, Alexa-568, and Alexa-647 were obtained from Molecular Probes/Life Technologies, UK and Jackson ImmunoResearch Laboratories Inc, West Grove, PA.

### Molecular biology and protein purification

The libraries of hexahistidine-GST in pFAT2 and eGFP-tagged Rab GTPases and human GEF coding sequences have been described previously (*Yoshimura et al., 2010*). Mutagenesis was performed using the Quickchange method according to the protocol (Agilent Technologies, UK). Rab proteins in pFAT2 were expressed in BL21 (DE3) pRIL at 18°C for 12–14 hr, then purified using Ni-NTA agarose as described previously (*Fuchs et al., 2007*). In brief, cell pellets were lysed in 20 ml IMAC20 (20 mM Tris–HCl, pH 8.0, 300 mM NaCl, 20 mM imidazole, and protease inhibitor cocktail; Roche Diagnostics, UK) using an Emulsiflex C-5 system (AVESTIN, Germany). Lysates were clarified by centrifugation at 16,000×*g* rpm in a JA-17 rotor for 30 min. To purify the tagged protein, 0.5 ml of nickel-charged NTA-agarose (QIAGEN, UK) was added to the clarified lysate and rotated for 2 hr. The agarose was washed three times with IMAC20 and the bound proteins eluted in IMAC200 (IMAC20 with 200 mM imidazole) collecting 1.5 ml fractions. All manipulations were performed on ice or in an 8°C cold room. Hexahistidine-tagged Rabex5, DENND1B-S, in pQE32 were expressed in JM109 at 18°C for 12–14 hr, then purified using nickel-charged NTA agarose using the same procedure as the Rabs. RabGAPs RUTBC3, TBC1D10A and TBC1D20 were purified as described previously (*Fuchs et al., 2007*). Purified proteins were dialyzed against TBS (50 mM Tris–HCl, pH 7.4, and 150 mM NaCl) and then snap frozen in liquid nitrogen

for storage at −80°C. Protein concentration was measured using the Bradford assay. TRAPPI-complex and DrrA were purified as described elsewhere (*Cai et al., 2008*; *Schoebel et al., 2009*).

## Nucleotide binding and Rab GEF assays

First 10 nmol of hexahistidine-GST-Rab was loaded with 2′-(3′)-bis-*O*-(*N*-methylanthraniloyl)-GDP (Mant-GDP) (Jena Bioscience, Germany) in 20 mM HEPES, pH 6.8, 1 mg/ml BSA, 20 mM EDTA, pH 8.0, 40 mM Mant-GDP at 30°C for 30 min. After loading 25 mmol $MgCl_2$ was added and the sample was exchanged into reaction buffer (20 mM HEPES, pH 6.8, 1 mg/ml BSA, 150 mM NaCl, 1 mM $MgCl_2$) using Zeba spin columns (Fisher Scientific). This step removes the free Mant-GDP leaving only Rab bound nucleotide. Nucleotide exchange was then measured using 1 nmol of the loaded Rab and the amount of GEF specified in the figure legends in a final volume of 100 µl reaction buffer by monitoring the quenching of fluorescence after release of Mant-GDP using a Tristar LB 941 plate reader (Berthold Technologies, UK) under control of MikroWin Software. Samples were excited at 350 nm and emission monitored at 440 nm. GTP was added to a final concentration of 0.1 mM to start the exchange reaction at 30°C. Curve fitting and extraction of pseudo first order rate constants ($k_{obs}$) was carried out as described previously (*Delprato et al., 2004*; *Delprato and Lambright, 2007*). Since $k_{obs} = (k_{cat}/K_m)$ [GEF] + $k_{basal}$ where $k_{basal}$ is the rate constant measured in the absence of GEF, catalytic efficiency ($k_{cat}/K_m$) can be obtained.

## Rab GAP assays

For GTP-loading reactions, 10 µl assay buffer (20 mM HEPES, pH 6.8, 1 mg/ml BSA), 73 µl $H_2O$, 10 µl of 10 mM EDTA, pH 8.0, 5 µl of 1 mM GTP, 2 µl γ-[$^{32}$P]-GTP (6000 Ci/mmol 10 mCi/ml Perkin Elmer, UK), and 100 pmol Rab-GTPase were mixed on ice. After 30 min of incubation at 30°C, loaded GTPases were stored on ice. GAP reactions were started by the addition of GAP to 1 µM Rab in 50 µl final volume as specified in the figures. A 2.5 µl aliquot of the assay mix was scintillation counted to measure the specific activity in counts per minute per picomole of GTP. Reactions were then incubated at 30°C for 0 to 60 min and then split into two equal aliquots. 5 µl of each aliquot was immediately added to 795 µl of ice-cold 5% (wt/vol) activated charcoal slurry in 50 mM $NaH_2PO_4$, left for 1 hr on ice, and centrifuged at 20,000×*g* to pellet the charcoal. A 400 µl aliquot of the supernatant was scintillation counted, and the amount of GTP hydrolyzed was calculated from the specific activity of the reaction mixture.

## Shiga toxin uptake assays

HeLa cells were cultured on No. 1.5 glass coverslips (Menzel-Gläser, Fisher Scientific) in DMEM containing 10% bovine calf serum (Invitrogen) at 37°C and 5% $CO_2$. Endogenous Rab35 was depleted using siRNA duplexes obtained from Qiagen directed against the 3′-UTR (Hs_RAB35_4 SI00092638), target sequence 5′-CCTGGGAAGAACCGAGTTTAA-3′ transfected using Oligofectamine (Life Technologies) for 72 hr. Cells were then transfected with eGFP-Rab35 or Rab35$^{Q67A}$ for 18 hr using Mirus LT1 (Mirus Bio LLC, Madison, WI). Shiga toxin assays were then carried out as described previously (*Fuchs et al., 2007*). For imaging samples were washed twice with 2 ml of PBS, and fixed with 2 ml of 3% (wt/vol) paraformaldehyde in PBS for 15 min. Fixative was removed and the cells quenched with 2 ml of 50 mM $NH_4Cl$ in PBS for 10 min. Coverslips were washed three times in 2 ml PBS before permeabilization in 0.2% (vol/vol) Triton-X 100 for 5 min. Cells were then stained with GM130 antibodies. Primary and secondary antibody staining was carried out in PBS for 60 min at room temperature. Coverslips were mounted in Mowiol 4-88 mounting medium (Merck Millipore, UK). Fixed samples on glass slides were imaged using a 60x NA1.35 oil immersion objective on an Olympus BX61 upright microscope with filter sets for DAPI, GFP/Alexa-488, Alexa-555, Alexa-568, and Alexa-647 (Chroma Technology Corp., Bellows Falls, VT), a CoolSNAP HQ2 camera (Roper Scientific), and Metamorph 7.5 imaging software (Molecular Dynamics Inc., Sunnyvale, CA).

## FRAP and membrane fractionation assays for Rab recruitment

For live cell imaging using spinning disk confocal microscopy, cells were plated in 35 mm dishes with a 14 mm No. 1.5 thickness coverglass window in the bottom (MatTek Corporation, Ashland, MA). Cells were left for 24 hr then transfected with eGFP-Rab constructs for a further 16 hr. For imaging, the dishes were placed in a 37°C and 5% $CO_2$ environment chamber (Tokai Hit CO., Ltd, Japan) on the microscope stage. Imaging was performed at 37°C in 5% $CO_2$ using an Olympus IX81 inverted microscope with a 60x 1.42NA oil immersion objective coupled to an Ultraview Vox spinning disk confocal system (Perkin Elmer) fitted with a C9100-13 EM-CCD camera (Hamamatsu Photonics Limited, UK).

For FRAP, five image stacks of four planes with 0.2 µm spacing were acquired at 1 s intervals during the pre-bleach period. Bleach of the eGFP-Rab signal was performed using an UltraVIEW PK Device with the 488 nm laser set at 10% with the following settings: cycles 5, step size 1, spot period 10, stop period 10, spot cycles 1, small spot size and no attenuation. Recovery images were acquired for 30 time points every 2 s using 50 ms exposures at 4% laser power and then a further 30 time points every 4 s. Quantification and analysis of the FRAP data were performed using ImageJ. For membrane fractionation the cells were washed from the dish in PBS containing 1 mM EDTA, then homogenized using 20 passes through an 18-gauge needle in 50 mM HEPES-NaOH pH7.4, 200 mM sucrose. Unbroken cells were removed by centrifugation at 1000×$g$ for 10 min in a microfuge. A membrane pellet and cytosol were prepared from this post-nuclear supernatant by centrifugation at 100,000×$g$ for 60 min in a TLA-100 rotor. Equivalent proportions of the membrane pellet and cytosol were analysed by western blotting.

## Additional information

### Funding

| Funder | Grant reference number | Author |
|---|---|---|
| Wellcome Trust | 097769/Z/11/Z | Francis A Barr |

The funder had no role in study design, data collection and interpretation, or the decision to submit the work for publication.

### Author contributions

LL, Acquisition of data, Analysis and interpretation of data, Drafting or revising the article; RNB, Acquisition of data, Analysis and interpretation of data, Contributed unpublished essential data or reagents; YC, AI, KR, Analysis and interpretation of data, Drafting or revising the article, Contributed unpublished essential data or reagents; FAB, Conception and design, Analysis and interpretation of data, Drafting or revising the article

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
