## [Decision Letter]

Thank you for sending your work entitled “Plasticity in nucleotide release mechanism results in coupling of Rab GTPase activation and inactivation” for consideration at *eLife*. Your article has been evaluated by a Senior editor and 3 reviewers, one of whom, Suzanne Pfeffer, is a member of our Board of Reviewing Editors.

The Reviewing editor and the other reviewers discussed their comments before we reached this decision, and the Reviewing editor has assembled the following comments to help you prepare a completely revised submission.

This is an important and interesting paper that shows that Rab GTPase GEFs and GAPs act by distinct mechanisms, even when a single Rab is the substrate. The work has the potential be of broad interest to cell biologists interested in molecular mechanisms. However, the manuscript needs significant reworking to clarify the story for the non-cognoscenti. The authors are encouraged to step back and rewrite the paper to guide the reader through the structures carefully, one figure at a time and keeping all of the referee comments in mind when crafting their presentation.

The reviewers felt strongly that the use of the word coupling is not correct and would be confusing to cell biologists. The concept of coupling implies coupled reactions, which is not the case since the GEFs and GAPs presumably act in different locations in cells. In addition, two expert enzymologists feel that classifying families based on one Q or D/E residue is not a sound practice. You need to convince them that this is helpful for the field if you believe otherwise. Finally, one wrote, “...The authors need to re-draw their conclusions in a way that maintains a high level of scholarly integrity.” I think they mean tone down the idea that this is a major conceptual advance – it is, nevertheless, a surprising and interesting result that these enzymes use different mechanisms. This reviewer felt a change in the manuscript title would also help.

Because rewriting is required I attach here all the referee comments. Normally, a complete rewriting could be cause for rejection but the referees believe that a very interesting finding can be found in the core of this manuscript and encourage a second round of evaluation.

*Reviewer #1*:

1) The figure legends need to describe the figures much more clearly and accurately. 1b has no legend; Figure 3 legend is out of order; Figure 4 are either reverse labeled or need better clarification in terms of what is presented.

2) Figure 2, panel b, bottom. Please show the Rab1-GDP with the same projection as Rab1-DrrA and/or move it for comparison with Rab1-GTP to indicate residue orientation for comparison. Is there a Q-GEF for Rab1 or is that a microbial adaptation and approach to perfection? This could be tested – does Rab1-Q mutant rescue a Rab1 phenotype.

3) Figure 3. Panel d should be moved to Figure 7b. Please compare Rab1-GTP with Rab1-GDP and show key residues for comparison. Bottom text “Divergent in Q/E positons” is misplaced.

4) Figure 4. Please include Bet3 E192 mutant for completeness. Can you include an S-GEF example? Please relabel b,d,f,h so that the reader understands what is shown.

5) Figure 5 basal hydrolysis needs to be shown as a rate, not a one hour time point. This is important since GTP hydrolysis rates vary significantly between Rabs. Would inclusion of the Rab5Q or Rab5GTP structure be useful to show here to understand why Rab5Q mutant is so poor in intrinsic hydrolysis?

6) How many cells were counted in 6c? The images would be easier to understand if cells expressing the particular Rab were indicated – it seems some cells express more of the Rab and the phenotype does not seem uniform. Please present this more clearly. Also, the FRAP in d, e may be the result of differences in capacity to interact with GDI and without monitoring relative prenylation and cytosol/membrane fraction cannot really be interpreted. Please remove or add additional data so that the FRAP can be understood molecularly.

Please include the fact that RUTBC1 was shown to activate a catalytically inhibited Rab33B mutant (Q92A), in support of a dual finger mechanism for RUTBC1 action (Nottingham & Barr et al. 2011).

*Reviewer #2*:

The manuscript by Langemeyer et al investigates the mechanisms by which different Rab GTPases are activated by the nucleotide exchange activity of their respective GEFs and, in some cases, whether this activation is 'coupled' to inactivation by the GAPs. Starting from a survey of available crystal structures and sequence information, the authors noted that different GEFs may use distinct residues (Q, E/D, or S, sometimes in the Rab Switch II loop and sometimes supplied by the GEF) for activation of Rab, and the same Rab (e.g., Rab1) may use distinct residues for activation by different GEFs. This is experimentally tested in the cases of Rab35 (Q), Rab5 (D), and Rab1 (Q- or E-) for GEF-mediated nucleotide exchange and GAP-mediated GTPase activation. These experiments also show that the same glutamine in Rab35 and Rab1 is important for both GEF-mediated activation and GAP-mediated inactivation, hence the notion of 'coupled' 'pathways' for activation and inactivation. The importance of the glutamine for activation of Rab35 in vivo is further verified in a Shiga toxin B uptake assay.

Overall, this is more of a 'hypothesis' paper (4 model/literature review figures vs 3 data figures) that (i) suggests a new way of classifying the Rab GEFs based on the residue used in Rab activation; and (ii) proposes that activation and inactivation of some Rabs that use Q-GEFs are coupled. The experiments are done well and made the points summarized above, but by themselves they are not highly novel if not placed in the context of the authors' hypotheses.

Regarding (i), the question that immediately arises is whether it is justified to classify Rab GEFs based on a single residue that participates in nucleotide exchange. I have a hard time doing so. The idea of a single residue being responsible for, and is hence the 'key' for activation, is incompatible with the principles of macromolecular activity, which is almost always the result of a collective network of interacting residues. For example, will Q67 explain GEF-induced displacement of Switch I loop and active site Mg2+? Are all other mechanisms of disrupting the Rab-GDP interaction the same for GEFs? If the authors cared to mutate other residues around the Rab or GEF active site, will they identify other mutations that disrupt GEF-mediated activation? (The answer is almost certainly yes.) Given this, what justifies placing paramount importance on the Q or D/E in Switch II of Rab? Second, such a classification offers no predictive power, as both Q and E residues are conserved in the Rab Switch II loop, but only one of them are used, and sometimes neither is used. Most researchers will have to begin with a null hypothesis with or without knowing that GEFs can be Q, D/E or S family. Thirdly, the available data are not sufficiently comprehensive to support, for example, the presence of the 'S-family' or that the Ypt1is a D/E family member.

Regarding (ii), while I agree that the same glutamine is used by Rab35 and Rab1 for GEF-mediated exchange and GAP-mediated hydrolysis, I have a hard time considering this as 'coupled' activation and inactivation. One can certainly identify many aspects of the mechanisms of these two reactions that are highly different, and hence 'uncoupled'. 'Coupled' in this paper has changed meaning, and refers to a requirement for the Switch II glutamine. To what extent is this concept valuable? Only when this glutamine is mutated, and one thus disrupt both sets of Rab regulation. Pragmatically, I agree with the authors that this gives researchers in the field a warning not to assume that mutating the Switch II glutamine only disrupts GAP activity and to make wrong interpretations, but that seems all there is – I can not come up with more examples in which this 'coupled' concept would be useful. I will be happy to hear more if the authors can explain.

[Minor comments not shown]

The title (Plasticity in nucleotide release mechanism results in coupling of Rab GTPase activation and inactivation) implies a causal relationship that I have a hard time digesting. Plasticity in this paper refers to Rabs being able to use Q, D/E, or S for activation by GEF. “Coupling of activation and inactivation” refers to the observation that for a Rab that uses a Q-GEF, the same Q can also be used in GAP-mediated hydrolysis. Do the authors see a strong connection between the two that I don't?

*Reviewer #3*:

The starting off point for this work are the structures of complexes between various Rab proteins and their cognate GEFs, in particular with reference to the fate of an essential lysine residue in the P-loop of the GTPases after displacement of GDP by a GEF. It had been suggested earlier that a common feauture of such GTPase:GEF complexes is an interaction of this lysine with a glutamate residue in the switch II region immediately following the catalytic glutamine. Comparing the structures discussed in the manuscript, it is clear that this does not apply generally, in particular here for the Rab proteins. The authors define 4 classes of GEFs, which they call the D, E, Q or S GEFs, where the single letter refers to the type of residue that interacts with the P-loop lysine in the Rab:GEF structure.

These residues can be in the switch II region, elsewhere in the Rab sequence or in the GEF. Of special interest, and this is the origin of the title of the manuscript and of several statements made throughout the manuscript to the effect that there is coupling between GTPase activation and inactivation, are the Q GEFs. The origin of this statement is the fact that in the Q GEFs, the lysine interacts with the conserved catalytic glutamine in the Rab:GEF complexes for DENND1 and DrrA, GEFs for Rab35 and Rab1, respectively. This is an interesting observation, and mutational analysis (referred to as alanine scanning analysis, which it doesn't appear to be) confirms that in these cases, this glutamine is important not only for GTPase activity, as already known, but also for GEF activity. However, and this is my main criticism of the manuscript, this has nothing to do with coupling of activation and inactivation of Rabs, whatever this is supposed to mean. The fact that the same glutamine residue is important for the GEF reaction and the GTPase reaction does not imply coupling in any manner. Thus, after displacement of GDP and generation of the Rab:GEF complex, there will be immediate binding of GTP and dissociation of the GEF, but what happens next, or better, the time scale of what happens next depends on the rate of the intrinsic (basal) GTPase reaction and the availability and activity of a cognate GAP, but this has nothing to do with the question of whether the P-loop lysine interacted with a glutamine, a glutamate, an aspartate or a serine in the nucleotide-free Rab:GEF complex, and where this residue is in which sequence.

The emphasis on the (in my opinion incorrect) idea of coupling of activation and inactivation means that the manuscript cannot be published in this form. The observations are interesting in terms of basic principles, since they show that there are apparently several ways of stabilizing Rab:GEF complexes via interactions with the P-loop lysine (in one case 2 different ways for the same Rab), but they are probably most interesting with respect to the use of mutations of the essential glutamine for cell biological studies. This approach to generating stable GTPase:GTP complexes for cell biological studies is known to be flawed in the case of Rabs because the effect on GAP activation is much less than for other GTPases because TBC domain containing GAPs supply a glutamine to take over the role that the switch II glutamine plays in other classes of GTPases. The work presented here demonstrates that the approach is flawed for a further reason, i.e., because in the case of the Q Rabs activation will be inhibited. This is important information for scientists working in this area, but is only relevant to the use of these mutants, and not to the physiological situation.

The authors need to reconsider what the main message of this paper should be. It cannot be that RabGTPase activation and inactivation are coupled for some classes of GEFs, or better Rab-GEF combinations.

On examining the structures discussed in a little more detail, there are several more points to be made. If the P-loop lysine interacts with a neutral sidechain (glutamine, serine), the question arises as to charge neutralization of the protonated lysine amino group. Looking at the Rab1:DrrA structure, it is clear that there is a strong interaction with D63 from the Rab molecule (reported in the Schoebel et al. paper in Mol.Cell, 2009), in fact probably much stronger than the one with with Q67 discussed in this manuscript, where there is a bond length of 3.6 Angstroms. D63is the highly conserved aspartate in the WDTAGQE sequence, and is in fact the equivalent residue to D74 in Rab5 that was identified in this manuscript (and earlier, of course) as the Rab residue interacting with the P-loop lysine in the complex with Vps9. Looking at the DENND1:Rab35 structure, we see the same constellation (i.e., interactions of the lysine with D63 and Q67). So should DrrA and DENND1 be called D/Q Rabs? And Rab5 a D/E Rab? In the case of Sec2:Sec4, examination of one structure in the pdb does indeed show an interaction of the lysine with Ser161. A quick look for an acidic residue for charge neutralization did not reveal an obvious partner, although D101 of Sec4 is quite near. The GEF molecule is too far away to interact. In another Sec2:Sec4 structure in the PDB, the lysine is far removed from Ser161, but interacts with an inorganic phosphate group bound to the complex. These are all points that need discussion in the manuscript.

The reader is left with the distinct impression that an attempt has been made to make a bigger story out of the results than is justified. This oversell is unworthy of the intelligence and reputation of the authors. Please rewrite with an emphasis on interpretations that are justified by the arguments.

[Minor comments not shown]

[Editors’ note: further clarifications were requested prior to acceptance, as described below.]

Thank you for resubmitting your work entitled “Diversity and plasticity in Rab GTPase nucleotide release mechanism has consequences for Rab activation and inactivation” for further consideration at eLife. Your revised article has been favorably evaluated by a Senior editor and me. All the reviewers agree that the manuscript has been improved significantly but there are some remaining issues that need to be addressed before acceptance, as outlined below:

Reviewer 2 wrote:

“Overall the clarity of the presentation is significantly improved from the previous manuscript. The attempt of the authors to downplay the 'classification' of RabGEFs into different families is also appreciated. The distinction between RabGEFs that stabilize the P-loop by directly coordinating the P-loop lysine or repositioning the switch II loop of Rab to coordinate this lysine may provide a conceptually sounder line of classification, although this still awaits to be seen.

Much more emphasis in this revision appears to be placed on the contribution of the Switch II glutamine to the intrinsic GTP hydrolysis rate of the Rabs. This raises questions about the quality of GTP hydrolysis measurements in Figure 3. I am puzzled by the data. First, the units reported denote mass (pmol), whereas a hydrolysis rate or rate constant should be reported in time units (s-1 or min-1). Secondly, the basal GTP hydrolysis activity is presumably the y-intercept in the main panels, but these values do not match those in the inset that reports 'basal GTP hydrolysis'. In Figure 3 the basal GTP hydrolysis activity is in fact higher than the GAP-activated hydrolysis. Thirdly, I would assume that a proper 'GTP hydrolysis' measurement involves a time course (as all the arguments are about kinetics of hydrolysis), but re-reading the Methods section, I am less sure this time how these hydrolysis rate constants were measured and analyzed. PLEASE CLARIFY. Even if all the measurements are accurate, the author's statement “ in cells basal hydrolysis is a more significant determinant of Rab5 lifetime and turnover than GAP-mediated turnover” is highly speculative. The qualifier for this sentence “Together these findings indicate that...” needs to be significantly softened.”

[Minor comments not shown]

Roger Goody wrote:

“I think there is still more to be done on the role of the conserved aspartate in the Rab1-DrrA and Rab35-DENND1 interaction. As I alluded to in previous comments, mutation of this aspartate is problematic, since it has a negative effect on nucleotide binding, making the interpretation of the results difficult. Thus, the question arises as to whether we should be comparing kcat/Km values for wt and mutant, or perhaps the acceleration of the GDP release rate at saturating GEF concentrations. From Figure 2, it looks as if the rate constant for uncatalyzed GDP release from Rab1b with the aspartate removed is ca. 0.001/s, about 100 times faster than for wt-Rab1b. Thus, what is interpreted here as an increase of efficiency of GDP release catalyzed by DrrA on removing the aspartate might turn out to be a reduction if the acceleration ratio is considered. In addition, there is still the question of how charge neutralization of the P-loop lysine occurs without a negatively charged interaction partner, even if, as the authors point out, it is not clear whether the lysine is protonated or not. To my knowledge, without having checked exhaustively, there is always such an interaction in binary GTPase-GEF complexes, either in cis or trans (apart from the relatively poorly defined Sec2:Sec4 structure that has fortunately now been removed from the discussion in the present paper). I wonder whether there might be a change of mechanism with the aspartate mutant? It would be interesting to see which residues the lysine interacts with in the binary complex when the conserved aspartate has been removed. However, I am not suggesting that this structure should be solved before acceptance of this paper.”

* The Reviewing editor would be satisfied with a careful discussion of this point without a need for additional work. *

[Minor comments not shown]

---

## [Author Response]

[Editors’ note: the author responses to the first round of peer review follow.]

Reviewer #1:

*1) The figure legends need to describe the figures much more clearly and accurately. 1b has no legend;*
Figure 3
*legend is out of order;*
Figure 4
*are either reverse labeled or need better clarification in terms of what is presented*.

We have thoroughly revised the manuscript to more clearly present the data and explain the key findings. All figure legends and the main text have been revised. Figures and panels are now labeled strictly in the order of appearance.

*2)*
Figure 2*, panel b, bottom. Please show the Rab1-GDP with the same projection as Rab1-DrrA and/or move it for comparison with Rab1-GTP to indicate residue orientation for comparison. Is there a Q-GEF for Rab1 or is that a microbial adaptation and approach to perfection? This could be tested – does Rab1-Q mutant rescue a Rab1 phenotype*.

Structures are now are presented in the new Figure 1. We have spent some time trialing different figure layouts and hope this is clearer than the previous representation. The Rab1-GDP structure was removed as part of these changes in response to the comment of Reviewer 2 that some aspects of the figures were too much like a literature review.

*3)*
Figure 3*. Panel d should be moved to Figure 7b. Please compare Rab1-GTP with Rab1-GDP and show key residues for comparison. Bottom text “Divergent in Q/E positons” is misplaced*.

This figure panel was removed in the revision.

*4)*
Figure 4*. Please include Bet3 E192 mutant for completeness. Can you include an S-GEF example? Please relabel b,d,f,h so that the reader understands what is shown*.

The effects of TRAPP and Rabex acidic finger mutations are now tested, and additional mutations in the Rab switch II region are included as requested by the other referees. Together these data provide additional support the major conclusions of the work.

Because the manuscript has been extensively restructured the figure numbering has now changed. This data is now presented more simply in Figure 2.

*5)*
Figure 5
*basal hydrolysis needs to be shown as a rate, not a one hour time point. This is important since GTP hydrolysis rates vary significantly between Rabs. Would inclusion of the Rab5Q or Rab5GTP structure be useful to show here to understand why Rab5Q mutant is so poor in intrinsic hydrolysis*?

The values shown are the amount of GTP hydrolysis per hour under conditions that allow the effect of the GAP to be tested. This is within the experimentally validated linear range of the reaction and the value is therefore a rate (pmol/h).

*6) How many cells were counted in 6c? The images would be easier to understand if cells expressing the particular Rab were indicated – it seems some cells express more of the Rab and the phenotype does not seem uniform. Please present this more clearly*.

Over 300 cells for each of two independent experiments were counted for the Shiga toxin transport assays. The wild type rescue and mutant phenotypes were highly reproducible, and non-transfected cells depleted of Rab35 did not transport Shiga toxin into the Golgi apparatus. Cells rescued with the switch II mutant Rab35 did not support Shiga toxin traffic into the Golgi, and it is spread throughout the cytoplasm in what we think are small vesicles. Cells transfected with the Rab are outlined to focus the attention of the reader, and we thank the referee for this suggestion.

*Also, the FRAP in d, e may be the result of differences in capacity to interact with GDI and without monitoring relative prenylation and cytosol/membrane fraction cannot really be interpreted. Please remove or add additional data so that the FRAP can be understood molecularly*.

This is an important control and we have added a Western blot analysis of Rab cytosol/membrane distribution as suggested to address the issue of whether prenylation is altered by the mutations. No differences were seen, supporting the original interpretation of the FRAP data.

*Please include the fact that RUTBC1 was shown to activate a catalytically inhibited Rab33B mutant (Q92A), in support of a dual finger mechanism for RUTBC1 action (Nottingham & Barr et al. 2011)*.

This reference is now added and we apologize for this omission.

Reviewer #2:

*[…] The idea of a single residue being responsible for, and is hence the 'key' for activation, is incompatible with the principles of macromolecular activity, which is almost always the result of a collective network of interacting residues. For example, will Q67 explain GEF-induced displacement of Switch I loop and active site Mg2+? Are all other mechanisms of disrupting the Rab-GDP interaction the same for GEFs? If the authors cared to mutate other residues around the Rab or GEF active site, will they identify other mutations that disrupt GEF-mediated activation? (The answer is almost certainly yes.) Given this, what justifies placing paramount importance on the Q or D/E in Switch II of Rab*?

We thank the referee for the comments that the experiments are well done, but don’t completely agree with the view that this is only a question of GEF classification. Both this reviewer and Reviewer 3 acknowledge that questions remain about the mechanisms used by Rab GEFs. We show experimental data confirming that there is plasticity in terms of the P-loop switch II conformations leading to GDP-release. It is of course a conformation induced by GEF binding, and this is perhaps the crux of the matter. A single Rab is potentially permissive for multiple different conformations, and these are in fact used as the analysis of Rab1 with TRAPP and DrrA shows. This is an important piece of information for anyone studying GTPases and their function.

A further point is that we have identified mutations that discriminate nucleotide exchange from nucleotide binding. Previous work has shown that conserved aromatic residues in switch I are important for nucleotide exchange; however this is because they contribute to nucleotide binding as the referee notes. Mutation of these aromatic residues, for example, increases basal nucleotide release. The switch II glutamine mutations characterized here for DENN and DrrA GEFs show unaltered nucleotide binding properties, and reduce GEF-stimulated nucleotide release. We don’t disagree with the view that a complex network of interactions promotes nucleotide release, however we feel this work identifies a key node in this network – the switch II P-loop lysine interaction.

*Second, such a classification offers no predictive power, as both Q and E residues are conserved in the Rab Switch II loop, but only one of them are used, and sometimes neither is used. Most researchers will have to begin with a null hypothesis with or without knowing that GEFs can be Q, D/E or S family*.

Our analysis shows that DENND1 a member of the large (27) group of DENN GEFs acts via a switch II glutamine (Q) dependent mechanism. Therefore, the switch II Q mutation should probably be treated with caution for this family. Although the intrinsic and GAP-mediated GTP-hydrolysis rates are likely to be decreased by this oft-used mutation, the Q67A (and homologous) substitutions will also affect the activation of Rabs by DENN/DrrA GEFs. Consequently, the activations state of such Rab mutants will be ill defined under physiological conditions, as shown by our DENND1-Rab35 data in vitro and in vivo. Different to Ras, the switch II glutamate (E) is never used in Rabs so far as we can tell. The switch II aspartate (D) is used for Rabex, a member of the large group (∼11 in humans) of Vps9 domain GEFs. This isn’t something applying to only a few Rabs, since the data we provide is relevant for 30-40 human Rab GEFs and their target GTPases. That implies it has general value for future research.

*Thirdly, the available data are not sufficiently comprehensive to support, for example, the presence of the 'S-family' or that the Ypt1is a D/E family member*.

The original idea was to treat the Rab activation pathway as a journey, with the route referred to by a key waypoint along the journey. In the naming system we adopted this waypoint or intermediate was named according to the switch II interaction with the P-loop. In response to the comments from two referees who felt that this was a bad idea, we have revised the main text and figures to remove this naming convention, eliminate the model figures, and focus on the new data.

We accept the reservations about the electron density in the P-loop of the Sec2-Sec4 structure expressed by the referee; however we feel that this must still fall into a different class in terms of its activation mechanism. To test this we have measured GEF activity of human Rabin8 towards Rab8 and find catalytic efficiency of 4.2x10^4^ M^-1^ sec^-1^ higher than the published value of 2.6x10^4^ M^-1^ sec^-1^ (Figure 6; Guo et al. JBC 288:32466-32474).Author response image 1.

Here we have used full-length Rab8 and Rabin8 rather than partly truncated proteins, which may explain this difference. Mutation of the switch II glutamine to alanine increased this further to 7.5x10^4^ M^-1^ sec^-1^, while mutation of T150 (equivalent to S161 in Sec4) resulted in loss of GEF activation. We have provided them to the referees to indicate that we have investigated this issue further and not simply ignored the comment.

Additional data has been added for all Rabs showing the effects of mutating the most highly conserved residues in switch II (see revised Figure 2).

*Regarding (ii), while I agree that the same glutamine is used by Rab35 and Rab1 for GEF-mediated exchange and GAP-mediated hydrolysis, I have a hard time considering this as 'coupled' activation and inactivation. One can certainly identify many aspects of the mechanisms of these two reactions that are highly different, and hence 'uncoupled'. 'Coupled' in this paper has changed meaning, and refers to a requirement for the Switch II glutamine. To what extent is this concept valuable? Only when this glutamine is mutated, and one thus disrupt both sets of Rab regulation. Pragmatically, I agree with the authors that this gives researchers in the field a warning not to assume that mutating the Switch II glutamine only disrupts GAP activity and to make wrong interpretations, but that seems all there is – I can not come up with more examples in which this 'coupled' concept would be useful. I will be happy to hear more if the authors can explain*.

The finding that Rab GTPase activation (GDP-GTP exchange) and inactivation (GTP hydrolysis) are linked in the way described is a previously unreported finding. Its value is that it adds to our detailed understanding of Rab regulation, and as the referee notes also highlights precautions that should be taken when mutating the switch II region. We also show that a single Rab can undergo activation via intermediates with different P-loop-switch II conformations. GEFs binding to different sites at the surface of the target GTPase do not necessarily have to induce the same conformational changes in switch I/II to promote nucleotide exchange. As we show, more than one P-loop-switch II conformation is permitted for a single Rab GTPase. This plasticity in the Rab family is surely a finding likely to have general relevance for the Ras superfamily, and should be valuable for future work.

[Minor comments and responses not shown]

*The title (Plasticity in nucleotide release mechanism results in coupling of Rab GTPase activation and inactivation) implies a causal relationship that I have a hard time digesting. Plasticity in this paper refers to Rabs being able to use Q, D/E, or S for activation by GEF. “Coupling of activation and inactivation” refers to the observation that for a Rab that uses a Q-GEF, the same Q can also be used in GAP-mediated hydrolysis. Do the authors see a strong connection between the two that I don't*?

We have altered the title to address this point. We show that plasticity in Rab GTPase nucleotide release mechanism has consequences for both activation and inactivation pathways, and the cellular function of Rabs.

Reviewer #3:

*[…] These residues can be in the switch II region, elsewhere in the Rab sequence or in the GEF. Of special interest, and this is the origin of the title of the manuscript and of several statements made throughout the manuscript to the effect that there is coupling between GTPase activation and inactivation, are the Q GEFs. The origin of this statement is the fact that in the Q GEFs, the lysine interacts with the conserved catalytic glutamine in the Rab:GEF complexes for DENND1 and DrrA, GEFs for Rab35 and Rab1, respectively. This is an interesting observation, and mutational analysis (referred to as alanine scanning analysis, which it doesn't appear to be) confirms that in these cases, this glutamine is important not only for GTPase activity, as already known, but also for GEF activity. However, and this is my main criticism of the manuscript, this has nothing to do with coupling of activation and inactivation of Rabs, whatever this is supposed to mean. The fact that the same glutamine residue is important for the GEF reaction and the GTPase reaction does not imply coupling in any manner. Thus, after displacement of GDP and generation of the Rab:GEF complex, there will be immediate binding of GTP and dissociation of the GEF, but what happens next, or better, the time scale of what happens next depends on the rate of the intrinsic (basal) GTPase reaction and the availability and activity of a cognate GAP, but this has nothing to do with the question of whether the P-loop lysine interacted with a glutamine, a glutamate, an aspartate or a serine in the nucleotide-free Rab:GEF complex, and where this residue is in which sequence*.

We have revised the figures and manuscript text to remove the two issues identified by the referee. Rather than refer to D/E/Q/S GEFs we simply describe the results for the different Rab and GEF combinations and discuss these to develop a more coherent picture that we hope will inform future work. The notion of coupling has been removed entirely and we now note that Rab activation and inactivation can share a common determinant with regard to the role of the switch II glutamine. This is likely to apply to the large (27 member in humans) family of DENN GEFs, and therefore has relevance for many trafficking pathways.

*The emphasis on the (in my opinion incorrect) idea of coupling of activation and inactivation means that the manuscript cannot be published in this form. The observations are interesting in terms of basic principles, since they show that there are apparently several ways of stabilizing Rab:GEF complexes via interactions with the P-loop lysine (in one case 2 different ways for the same Rab), but they are probably most interesting with respect to the use of mutations of the essential glutamine for cell biological studies. This approach to generating stable GTPase:GTP complexes for cell biological studies is known to be flawed in the case of Rabs because the effect on GAP activation is much less than for other GTPases because TBC domain containing GAPs supply a glutamine to take over the role that the switch II glutamine plays in other classes of GTPases. The work presented here demonstrates that the approach is flawed for a further reason, i.e., because in the case of the Q Rabs activation will be inhibited. This is important information for scientists working in this area, but is only relevant to the use of these mutants, and not to the physiological situation*.

Mechanistic details and detailed X-ray structures do count for the physiological situation, and are not only relevant for designing mutations to use in cell biological studies. This study is part of a larger body of work by many groups that aims to explain Rab GTPase regulation and function. We hope the referee will agree that without such details our understanding remains little better than cartoon models.

*The authors need to reconsider what the main message of this paper should be. It cannot be that RabGTPase activation and inactivation are coupled for some classes of GEFs, or better Rab-GEF combinations*.

We have thought carefully about the criticisms of all the referees, and have focused and simplified the manuscript as a consequence. The main message of the paper is focused on the role of the switch II region in Rab activation by different GEFs, and depending on the nature of the cognate GEF, the plasticity that emerges from this.

*On examining the structures discussed in a little more detail, there are several more points to be made. If the P-loop lysine interacts with a neutral sidechain (glutamine, serine), the question arises as to charge neutralization of the protonated lysine amino group. Looking at the Rab1:DrrA structure, it is clear that there is a strong interaction with D63 from the Rab molecule (reported in the Schoebel et al. paper in Mol.Cell, 2009), in fact probably much stronger than the one with with Q67 discussed in this manuscript, where there is a bond length of 3.6 Angstroms. D63is the highly conserved aspartate in the WDTAGQE sequence, and is in fact the equivalent residue to D74 in Rab5 that was identified in this manuscript (and earlier, of course) as the Rab residue interacting with the P-loop lysine in the complex with Vps9. Looking at the DENND1:Rab35 structure, we see the same constellation (i.e., interactions of the lysine with D63 and Q67). So should DrrA and DENND1 be called D/Q Rabs? And Rab5 a D/E Rab? In the case of Sec2:Sec4, examination of one structure in the pdb does indeed show an interaction of the lysine with Ser161. A quick look for an acidic residue for charge neutralization did not reveal an obvious partner, although D101 of Sec4 is quite near. The GEF molecule is too far away to interact. In another Sec2:Sec4 structure in the PDB, the lysine is far removed from Ser161, but interacts with an inorganic phosphate group bound to the complex. These are all points that need discussion in the manuscript*.

As the referee suggests, the ionic aspartate-lysine interaction may be stronger than the H-bridge between lysine and glutamine, but nobody knows whether the lysine in the Rab-GEF-complex is protonated or not. The protonation state will be important to estimate any strength of interaction. However, the crystal structures do not provide any data on this issue and it is therefore hard to come to any firm conclusion about the strength of interactions of the participating amino acid side chains. Still, the glutamine is important because the experiments we present clearly show this for DENN and DrrA GEFs. The glutamine is therefore involved either in stabilizing the lysine directly or in organizing its stabilization. The discrete distances in the various complexes don't resolve this issue unambiguously since they only provide a static snapshot. The argument we put forward is that the glutamine is located at the heart of the nucleotide-Rab-interaction in the case of DrrA- and DENND1-complexes and therefore contributes to the exchange reaction. It has been shown in other instances that the lysine requires stabilization during the transition through the nucleotide-free GTPase-GEF complex and therefore amino acid residues contributing to this stabilization must be important. We have also tested the role of the switch II aspartate in GDP-release for all the Rab-GEF pairs tested (Figure 2). Only in the case of Rabex is it required for GEF stimulated GDP-release. For the other Rabs it either results in an increase in basal release, consistent with the role in nucleotide binding. This suggests that the aspartate does not play an essential role in lysine stabilization during the GEF stimulated GDP-release reaction.

*The reader is left with the distinct impression that an attempt has been made to make a bigger story out of the results than is justified. This oversell is unworthy of the intelligence and reputation of the authors. Please rewrite with an emphasis on interpretations that are justified by the arguments*.

We have carried out a careful analysis of how Rab activation works, and generated data that we were very excited about. The referee has explained why they do not agree with all our ideas and the way they were presented. We have thought about this carefully and done our best to address their comments and revise some of our own conclusions in light of the valuable comments provided.

[Editors’ note: the author responses to the re-review follow.]

*Reviewer 2 wrote*:

*“Overall the clarity of the presentation is significantly improved from the previous manuscript. The attempt of the authors to downplay the 'classification' of RabGEFs into different families is also appreciated. The distinction between RabGEFs that stabilize the P-loop by directly coordinating the P-loop lysine or repositioning the switch II loop of Rab to coordinate this lysine may provide a conceptually sounder line of classification, although this still awaits to be seen*.

We agree with the comment of the reviewer, and would like to add that far more structures of Rab-RabGEF pairs are needed before we can make any definitive statements on GEF classification. It is clear that major questions about Rab regulation and function still need to be addressed.

*Much more emphasis in this revision appears to be placed on the contribution of the Switch II glutamine to the intrinsic GTP hydrolysis rate of the Rabs. This raises questions about the quality of GTP hydrolysis measurements in*
Figure 3*. I am puzzled by the data. First, the units reported denote mass (pmol), whereas a hydrolysis rate or rate constant should be reported in time units (s-1 or min-1). Secondly, the basal GTP hydrolysis activity is presumably the y-intercept in the main panels, but these values do not match those in the inset that reports 'basal GTP hydrolysis'. In*
Figure 3
*the basal GTP hydrolysis activity is in fact higher than the GAP-activated hydrolysis. Thirdly, I would assume that a proper 'GTP hydrolysis' measurement involves a time course (as all the arguments are about kinetics of hydrolysis), but re-reading the Methods section, I am less sure this time how these hydrolysis rate constants were measured and analyzed. PLEASE CLARIFY. Even if all the measurements are accurate, the author's statement “ in cells basal hydrolysis is a more significant determinant of Rab5 lifetime and turnover than GAP-mediated turnover” is highly speculative. The qualifier for this sentence “Together these findings indicate that...” needs to be significantly softened.*”

The referee asks an important question about the data plotted in Figure 3. GAP-stimulated and basal nucleotide hydrolysis for three Rab-RabGAP pairs are shown separately in this figure. This allows a simple comparison of the effects of the Rab switch II mutations on GAP-stimulated and basal hydrolysis. Basal hydrolysis is plotted in the bar graph in pmol/h. GAP-stimulated hydrolysis is plotted as a function of GAP concentration for wild type and switch II glutamine mutant Rabs. These values were measured using the protocol in the methods, and are plotted in the line graph. Hydrolysis is in pmol/h, and this is now correctly indicated in the figure. All GAP-stimulated hydrolysis values are corrected for basal GTP hydrolysis in the absence of GAP, so the y-intercept is not the basal hydrolysis value.

In response to this comment the text has been edited to soften the conclusions, and part of the text was moved to the discussion section. All the data shown was in the original submission, and relatively minor changes were made to the text describing these data. It is therefore unclear to us why the referee feels that more emphasis is placed on the contribution of the Switch II glutamine to the intrinsic GTP hydrolysis rate of the Rabs. We hope the changes we have made address the concern of the referee.

[Minor comments and responses not shown]

*Roger Goody wrote*:

*“I think there is still more to be done on the role of the conserved aspartate in the Rab1-DrrA and Rab35-DENND1 interaction. As I alluded to in previous comments, mutation of this aspartate is problematic, since it has a negative effect on nucleotide binding, making the interpretation of the results difficult. Thus, the question arises as to whether we should be comparing kcat/Km values for wt and mutant, or perhaps the acceleration of the GDP release rate at saturating GEF concentrations. From*
Figure 2*, it looks as if the rate constant for uncatalyzed GDP release from Rab1b with the aspartate removed is ca. 0.001/s, about 100 times faster than for wt-Rab1b. Thus, what is interpreted here as an increase of efficiency of GDP release catalyzed by DrrA on removing the aspartate might turn out to be a reduction if the acceleration ratio is considered. In addition, there is still the question of how charge neutralization of the P-loop lysine occurs without a negatively charged interaction partner, even if, as the authors point out, it is not clear whether the lysine is protonated or not. To my knowledge, without having checked exhaustively, there is always such an interaction in binary GTPase-GEF complexes, either in cis or trans (apart from the relatively poorly defined Sec2:Sec4 structure that has fortunately now been removed from the discussion in the present paper). I wonder whether there might be a change of mechanism with the aspartate mutant? It would be interesting to see which residues the lysine interacts with in the binary complex when the conserved aspartate has been removed. However, I am not suggesting that this structure should be solved before acceptance of this paper.*”

We briefly explain that the conserved aspartate in switch II has a role in nucleotide binding. This is indirect and the aspartate does not make direct contact with the bound metal ion. For example, in Rab1 the P-loop serine (S22) and threonine (T40) in the (TIGVD motif) make direct contact with the bound metal ion. In addition, the beta- and gamma-phosphates of the bound GTP contact the metal ion. All these interactions fall in one plane around the magnesium ion. Finally, there are two water molecules positioned above and below the metal ion. The aspartate (D63 in Rab1) may influence the environment and possibly contact of one of these water molecules as well as the P-loop serine (S22). The same water molecule is also predicted to interaction with the carbonyl oxygen of the polypeptide back at threonine 65 in the switch II region. Many studies of Ras superfamily GTPases use a serine to asparagine (S22N) mutation in the P-loop since this prevents magnesium binding, and hence stable GDP or GTP binding. Removal of the aspartate as we have done will reduce the affinity for Mg:GDP or GTP, but does not abolish binding in the same way.

We agree with the referee that the aspartate may play a role in charge neutralization of the P-loop lysine in some cases (DENND1-Rab35 and DrrA-Rab1). The discussion has been extended to mention this point.

We disagree with the use of an acceleration ratio as suggested by the referee. The biological activity of a GEF is only fulfilled if it produces a sufficient number of active Rab molecules in a given time. It is therefore the absolute amount of activity (Rab GTP created), not a ratio that is the relevant parameter. We were careful in wording the text to avoid making the statement made by the referee: “what is interpreted here as an increase of efficiency of GDP release catalyzed by DrrA”. Our text states that GDP release was increased and we don’t say this was because the GEF is more efficient. In fact, we suggest that this because removal of the aspartate contributes to nucleotide binding (via the coordination of the metal ion), and therefore limits GDP-release.

[Minor comments and responses not shown]